# OpenReview forum: "DeStein: Navigating Detoxification of Language Models via Universal Steering Pairs and Head-wise Activation Fusion"
_colmweb.org/COLM/2024/Conference — COLM_

### Official Review · Reviewer_qo2T · 2024-05-10

**Rating:** 6
**Confidence:** 3
**Ethics Flag:** 1

**Summary:**

The paper introduces DESTEIN, a novel method for reducing toxic outputs in language models by altering their internal representations in the activation space, using less resources and time. Empirical tests show that DESTEIN outperforms existing methods in detoxification efficacy while preserving output quality and diversity, and is scalable across multiple large language models.

**Questions To Authors:**

1. In section 3.2, the $P_{tox}^{i}$ and $C_{tox}^{I}$ is not clear, would be better to clarify these notations.
2. The meaning of l in equation 4 and 5 is different, one for instance and the other for layer. Would be better to change a notation.
3. In equation 4, is z computed for each layers or shared across all layers? And how do authors handle the token dimension or sequence length difference is not clear.
4. The linear classifier in linear probing is not clear, is it a single trainable matrix or heuristics classifier.

**Reasons To Accept:**

1. Strong experiment performance.

**Reasons To Reject:**

1. The selected models are all around 7B parameter size, which limits the versatility of the proposed method (from model parameter size perspective). Would be nice to include small (2B) and larger (13B) models.
2. Some technical details are not clear enough, see questions.
3. Using accuracy to determine weight is not very intuitive, would be nice to clarify motivation. And why need to plus 1 in equation 6. Last, is this weight change dynamically during inference or fixed after one run?
4. The method's generalization needs to be explored. Authors use train and test data from the same source, which share a similar distribution. In the real world, we might not have seen the toxic data before, which means the distribution of prompt input is very different from the distribution of steering pairs prompt. So it would be interesting to see the performance from different sources to show the generalization.

---

> ### Author Rebuttal · Authors · 2024-05-31
>
> Thank you for your constructive feedback! We will address your concerns and clarify any misunderstandings below.
>
> **Responses to the reasons to reject**
> - R1: Taking your advice,we added 1.3B and 13B models.The results show our method performs excellently on both.
>   |Model|Method|Exp.Max.Tox.|Tox.Prob.|Dist-2|Dist-3|PPL|
>   |-|-|-|-|-|-|-|
>   |GPT2-XL(1.3B)|base|0.56|0.59|0.85|0.847|18.142|
>   ||self-detoxify|0.425|0.32|0.877|0.866|72.00|
>   ||OURs|0.322|0.16|0.865|0.859|24.989|
>   |LLaMA2-13B|base|0.543|0.56|0.826|17.018|38.222|
>   ||self-detoxify|0.524|0.55|0.826|79.573|41.381|
>   ||OURs|0.353|0.19|0.835|20.252|39.405|
> - R2: We'll revise the necessary parts and provide clearer explanations. Please see the  “responses to the questions” section.
> - R3: We assume attention heads with higher accuracy have more toxicity-related information, so we give them higher weights during detoxification. We confirmed this with ablation experiments. The table below shows a significant difference between the top-n% and bottom-n%, proving the validity of using accuracy.
>   ||Exp.Max.Tox.|Tox.Prob.|PPL|
>   |-|-|-|-|
>   |bot-10%|0.433|0.37|40.714|
>   |top-10%|0.211|0.04|47.412|
>   |bot-50%|0.315|0.16|31.032|
>   |top-50%|0.262|0.10|33.163|
>
>   And,the value 1 is variable and is used to adjust the weight of this term.To avoid  confusion, we'll change it to $\delta$ and explain it.Last, the linear classifier's weights stay fixed after one run.
> - R4: Following your suggestion, we added the BOLD dataset, which differs from our original data sources.The results shows it performs well.Thank you for this valuable insight.
>   ||Exp.Max.Tox.|Tox.Prob.|Dist-2|Dist-3|PPL|
>   |-|-|-|-|-|-|
>   |base|0.276|0.071|0.865|0.844|26.999|
>   |goodtriever|0.192|0.01|0.825|0.825|32.292|
>   |dexperts|0.128|0.00|0.838|0.817|33.701|
>   |OURs|0.072|0.00|0.868|0.854|31.449|
>
> **Responses to the questions**
> - Q1: $P$ and $C$ in Section 3.2 represent prompt and completion.We'll clarify these notations in our revision.
> - Q2: Your feedback is greatly appreciated.We‘ll change the notation from "$l$" to "$m$" to represent instances.
> - Q3:$z$ is computed for each layer.For autoregressive LMs,we utilize the representations of the last token to represent the sequence because it captures the contextual information of the sequence.
> - Q4: The linear classifier is indeed a single trainable matrix.
>
> **Other**
> - Due to word limits,only key results are presented.Full results,analysis,and citations will be included in our revision. Thanks!

---

> > ### Comment · Reviewer_qo2T · 2024-05-31
> > **Thank you for your response!**
> >
> > Thanks authors for the detailed response, which address most of my concerns. I will raise my score accordingly.

---

> > > ### Author Response · Authors · 2024-06-01
> > >
> > > We're happy to know that our response helped in addressing your concerns. Thanks again for the helpful feedback and for upgrading the score!

---

### Official Review · Reviewer_1eer · 2024-05-11

**Rating:** 6
**Confidence:** 3
**Ethics Flag:** 1

**Summary:**

This paper presents a novel approach to detoxifying Language Models with detoxification vectors. By modifying LMs’ internal representations within the activation space, the proposed method effectively mitigates the need for increased inference time and computational resources. Experimental results across various LMs demonstrate the efficacy of the proposed approach in detoxifying LMs compared to existing methods.

**Questions To Authors:**

It seems that some variable descriptions are missing. (ex. $P^j_{tox}$ and $C^j_{tox}$ in Section 3.2)
I think it would be good if some examples of generated ($S_{tox}$, $S_{nontox}$) were shown.

**Reasons To Accept:**

The proposed method could detoxify LMs without increasing computational costs, and with minimal impacts on the model’s generative capabilities. Furthermore, the method's simplicity translates to fewer hyperparameters, facilitating easier tuning and implementation.

**Reasons To Reject:**

While the paper is well-written, it lacks comprehensive analysis to strengthen the validity of the proposed method.

The method's efficacy appears heavily reliant on the quantity and caliber of parallel steering pairs utilized, yet this aspect receives limited scrutiny in the paper. There appears to be an absence of analysis regarding the requisite amount of data and its quality for effective implementation. Furthermore, the paper seems to lack an investigation of the impact of varying the hyperparameters on the results. Examining these would provide insight into the robustness of the method and its sensitivity to parameter settings.

---

> ### Author Rebuttal · Authors · 2024-05-31
>
> We greatly appreciate your thoughtful review and feedback on our paper.We will address your concerns below.
>
> **Responses to the reasons to reject**
> - **R1.Comprehensive Analysis**
>   We have added two sets of ablation experiments to explore the impacts of activation fusion positions and probing-decided per-attention-head weights.Please refer to our response to "Reviewer bowP" in section "R1" and "Reviewer wsTL" in section "R2" for more details.
> - **R2.Quantity and Caliber of Parallel Pairs**
>   In Appendix E, we have included an analysis of the impact of the quantity of parallel steering pairs on the efficacy of our method. Based on your suggestion, we also examined the quality of parallel steering pairs and their effect on performance. The results are as follows.
>   ||Exp.Max.Tox.|Tox.Prob.|Dist-1|Dist-2|Dist-3|PPL|
>   |-|-|-|-|-|-|-|
>   |5-pairs|0.307|0.14|0.577|0.858|0.858|38.222|
>   |10-pairs|0.209|0.05|0.562|0.849|0.857|51.869|
>   |20-pairs|0.203|0.04|0.569|0.858|0.86|39.405|
>   |40-pairs|0.229|0.08|0.585|0.862|0.862|43.088|
>   |60-pairs|0.213|0.08|0.588|0.862|0.862|43.364|
>
>   The results show that for GPT2-L,selecting 20 parallel pairs was sufficient,as further increasing the number of pairs didn’t significantly improve performance. This balance between the quantity of pairs and performance guided our decision to use 20 pairs in our method.
>
> - **R3.Hyperparameter**
>   We systematically varied the hyperparameter $\alpha_{contr}$ and examined its impact on the results.In Section 5.1,we analyzed its influence using line charts.Now,we supplement this with tables for clearer presentation.
>   |$\alpha_{contr}$|Exp.Max.Tox.|Tox.Prob.|Dist-1|Dist-2|Dist-3|PPL|
>   |-|-|-|-|-|-|-|
>   |0.1|0.426|0.33|0.584|0.857|0.854|26.972|
>   |0.3|0.27|0.11|0.576|0.857|0.857|32.113|
>   |0.4|0.203|0.04|0.569|0.858|0.86|39.405|
>   |0.6|0.107|0.01|0.557|0.859|0.864|66.363|
>
>   The results indicate that the detoxification strength increases with the increase of $\alpha_{contr}$.The analysis revealed that the method is robust across a wide range of hyperparameter settings.
>
> **Responses to the questions**
> - Q1: $P$ and $C$ in Section 3.2 represent prompt and completion.We'll clarify these notations in our revision.
> - Q2: Examples of generated ($S_{tox}$,$S_{nontox}$) are provided in Appendix G.
>
> **Other**
> - Due to word limits,only key results are presented.Full results,analysis,and citations will be included in our revision. Thank you!

---

> > ### Comment · Reviewer_1eer · 2024-06-01
> > **Thank you for the response.**
> >
> > Thank you for your additional explanation. I now have a deeper understanding of how $\alpha_{contr}$ can control the detoxification strength. However, I am curious about how the generated text would change with a higher $\alpha_{contr}$ value (e.g., 0.6).
> > Overall, the paper is well-written, and I will maintain my current score.

---

> > ### Author Response · Authors · 2024-06-01
> >
> > Thank you for taking the time to review our paper and providing insightful feedback!
> > Furthermore, regarding your curiosity, in fact, when the $\alpha_{contr}$ is around 0.6, the generated texts are still fluent. The slight increase in PPL during the experiment is inevitable for the detoxification task because when facing toxic information in the prefix, the detoxified model has to do its best to reverse the situation, resulting in an increase in PPL. However, when the detoxification strength is excessively high, a collapse occurs, as analyzed in section 5.1 of our paper. This is predictable because we can't fully decouple toxicity, although we have made many improvements towards this goal.
> > Finally, thank you again for recognizing our work and for your insightful feedback.

---

### Official Review · Reviewer_wsTL · 2024-05-12

**Rating:** 7
**Confidence:** 4
**Ethics Flag:** 1

**Summary:**

This paper introduces an activation-intervention method for detoxification of language models. In particular, they (1) sample unconditionally from the target LM and filter for toxic examples, (2) use a much larger instruction-tuned model to produce non-toxic paired examples, (3) estimate differences in layerXattention_head activations between toxic and non-toxic examples, (4) estimate which (layer, attention head) best represents toxicity, (5) add in the non-toxic minus toxic activations average vector during inference to detoxify.

The paper explores various LMs, including GPT-2-large and LLAMA-2-7B. The results are strong in detoxification, and there is a LLM-based preference evaluation showing that their method’s outputs are preferred over other detoxification methods.

There are some interpretability claims in the paper (see below.)

**Questions To Authors:**

Please fix many of your citations, which do not have proper spacing.

Saying layer normalization isn’t crucial to your analysis is odd just before providing an equation for the value of a vector under a transformer at a layer and index, since the layer normalization is considered very important in that equation.

> Intuitively, to achieve detoxification without compromising the internal knowledge, the activation space at the output location of the MHSA should be the most optimal.

I don’t share this intuition. Can you explain?

When you say the activation space of each attention head, I’m confused – do you mean the model_dim//head_size dimensional vectors that are computed during MHA, but not passed through the rest of the model?

Is the probe really p(a_i^l) = sigmoid(a_i^l)? This both (1) includes no learnable parameters, and (2) has an output space of the dimensionality of a_i^l, not, e.g., 2 or 1, to allow for classification of toxic/non-toxic. Do you mean  p(a_i^l) = sigmoid(w^\top a_i^l) for some parameters w?

In equation 6, why add 1 to \alpha_\prob? As opposed to 0 or some other constant?

>  Nonetheless, this methodology, predicated on Probing for weight distribution, facilitates a heightened focus on activation spaces that exhibit a higher degree of disentanglement.

I disagree; just because the activation space is better at predicting toxicity does not mean it’s not also better at linearly predicting various other properties. In fact, many probing studies find that layers that best-encode one property often best-encode other properties as well. (e.g., Liu et al., 2019. Linguistic Knowledge and Transferability of Contextual Representations).

On page one you claim there’s a need for an “interpretable” method for detoxification – however, despite the brief experiments in 5.2, I don’t really see how this method is any more interpretable than, say, finetuning – and the claim isn’t really backed up, so I’d encourage you to remove that from page 1.

**Reasons To Accept:**

Overall, I find this paper to be a nice study in activation arithmetic, model intervention, and generating paired data that shows a simple recipe for a surprisingly powerful intervention to model behavior. The high-level experimental designs – model choices, additional experiments, baselines, etc – all make sense to me.

In reasons to reject, I’ve written quite a bit – there are a lot of lower-level (and some more important) details that make this paper weaker. However, my overall read is that the basic claims of the paper are well backed up by experiment, the method is relatively simple and useful, and the recipe provided might be of interest to a wide range of COLM attendees and general readers.

**Reasons To Reject:**

I’m concerned that PPL under GPT-2-XL is used to estimate fluency; this is not a very good model, and e.g., highly repetitive nonsensical text will achieve very low PPL under the model. But the inclusion of LLMs as judges does help alleviate this.

It’s unclear that the probing-decided per-attention-head weights are useful; the ablation in the appendix (as far as I can tell) doesn’t compare to the proposed method. Or if the two tables are comparable, then the fixed-to-0.5-ablation method seems pretty much the same as the proposed probing-based method. So, from the data provided, I don’t really see the head-wise weights as being very important to the method. A lot of space in the paper is given to this head-wise weighting through probing, so this is a bit of a red flag to me, but I’m not giving the paper a lower score because I’d even prefer if the no-probing method worked just as well, since it’s simpler. If the authors want to keep their arguments about disentanglement estimation through probing, though, I really think they need to make it clearer that it actually helps improve the method over a baseline.

Unfortunately, the authors try to use t-SNE to validate that there is a linear representation of toxicity within a representation space. This experiment, in which the post-tSNE space has linear separation, says nothing about linear separation in the original space, because tsne is a highly non-linear dimensionality reduction method. I strongly recommend that this claim be removed, or that the plot be replaced with a linear dimensionality reduction method, e.g., PCA, and linearity checked there.

The methods background 3.1 and methods description in 3.2,3.3 are too vague to get a precise understanding of what activations are being acted upon. (e.g., is it the head-level lower-dimensional vectors? I’d imagine so, but the stated math seems to suggest the full-dimensional token vectors before reshaping to dim//heads.)

The motivation of the linear representation hypothesis seems somewhat confused, especially given section 3.3, wherein the authors state that linearity is at best an approximation  due to “false correlations” or “biases” in the dataset, and that “entanglement of toxicity with unrelated attributes varies significantly across different activation spaces” all without citation or data to explain or provide evidence. There are a few instances of this kind of somewhat opaque paragraph in this paper that I find difficult to square with the otherwise straightforward method.

---

> ### Author Rebuttal · Authors · 2024-05-31
>
> We appreciate your thoughtful comments and suggestions,and we are glad to learn that you found our contribution valuable to the field!
>
> **Responses to the reasons to reject**
> - R1(PPL):Based on your suggestion, we retested the PPL with LLaMA3-8B. The results strengthen our methods.
>   ||Exp.Max.Tox.|PPL|
>   |-|-|-|
>   |GPT2-L|0.557|68.547|
>   |discup|0.300|85.290|
>   |OURs|0.203|83.104|
>
> - R2(weights):In Appendix C, the top-n and bottom-n differences show that accuracy can partially gauge the quality of attention heads. Weighting all heads slightly improved metrics, possibly due to PPL limits. Therefore, we introduced MMLU for further evaluation.
>   ||Ave.Acc.|
>   |-|-|
>   |GPT2-L|0.27|
>   |no-weight|0.255|
>   |weight|0.261|
>
>   The results show that the weighting method is effective. It is promising and may inspire future work. We plan to design more efficient fusion methods in the future.
> - R3(t-SNE):We'll replace the t-SNE plot with the PCA in our revision.
> - R4(description):We apologize for any confusion and will clarify the activations in our revision.
> - R5(motivation):Previous studies found a false correlation between toxicity and biases, especially racial bias. We think it varies in degree across different heads, so we use the racial group in the BOLD dataset to verify this.
>   |regard|$D_{pos}$\ $D_{neg}$|
>   |-|-|
>   |GPT2-L|2.64\3.74|
>   |bot-10|3.65\3.36|
>   |top-10|0.35\1.9|
>
>   The results show the bottom heads lead to a greater increase in bias,confirming our previous hypothesis.
>
> **Responses to the questions**
> - Q1-2(citations and layer normalization):We'll fix the citation spacing issues and revise the statements on the importance of layer normalization.
> - Q3(different activation):We added ablation experiments with activation fusion at FNN, MHSA, and FNN+MHSA. Please refer to our response to "Reviewer bowP" in section "R1".
> - Q4(attention head):Yes,it's the vectors that are computed during MHA.
> - Q5(probe):Yes,it's $p(a_i^l) = sigmoid(w^\top a_i^l)$.Thanks for noting.
> - Q6($\alpha_{prob}$):The value 1 is variable and is used to adjust the weight of this term. For our experiments, we chose 1. To avoid  confusion, we'll change it to $\delta$ and provide an explanation.
> - Q7:Thanks for your insightful feedback.We'll explore the disentanglement in the future.
> - Q8:Thanks! We‘ll remove "interpretable" claim from page 1.
>
> **Others**
> - Due to word limits, only key results are presented. Full results, analysis, and citations will be included in our revision.

---

> > ### Comment · Reviewer_wsTL · 2024-06-04
> > **Thank you; my positive review remains unchanged.**
> >
> > I thank the authors for their response and look forward to an improved paper. My recommendation to accept remains unchanged.
> >
> > I discourage the authors from using the MMLU scores to motivate their method. The baseline accuracy is 25%, and anything 27% or below is pretty reasonably just due to noise, so I don't think 27% vs 25.5% vs 26.1% are meaningfully different.

---

> > > ### Author Response · Authors · 2024-06-05
> > >
> > > Thank you very much for your positive feedback and for your valuable suggestions. We agree that your suggestion is reasonable. Given the limitations of GPT-2 L, using MMLU for evaluation may not be an appropriate choice. We will revise this part accordingly. Thank you once again for your time and consideration.

---

### Official Review · Reviewer_bowP · 2024-05-14

**Rating:** 7
**Confidence:** 4
**Ethics Flag:** 1

**Summary:**

Authors propose a method to ensure LM generated code is free of toxicity. This is done by computing a difference in the activations level of tokens from parallel toxic-nontoxic sequences, and adding this one set of universal detoxification vectors to activations during inference. This is shown to be effective in ensuring generation free of toxicity without hurting the linguistic quality (fluency, diversity) from LMs like GPT-2, Llama-2, MPT and OPT.

**Questions To Authors:**

Suggestions:
1. Sec 2, Para 2: Authors could also mention that fine-tuning methods can possibly lead to catastrophic forgetting (model performing poorer on the tasks it has seen before non-toxic fine-tuning).
2. Qualitative samples can help the reader to better understand the comparison between baselines and the proposed approach in Sec 4 or 5.

Questions:
1. Sec 2, Para 4: "The method we propose can also be categorized as a variant within the spectrum of prompt-based detoxification approaches" I did not follow this claim. The proposed method DESTEIN applies a shift by a pre-computed tensor to the activations in the transformer. How is this a prompt-based method then?
2. Sec 3.2, Para 2: "and its corresponding $P_{\text{tox}}^{i}$ and $C_{\text{tox}}^{i}$" - what are P and C denoting here? prompt and completion? 3.1 used the lower-case $p$ to denote prompts, so it's confusing for the reader to follow this. $p$ is again used in 3.3 to denote the sigmoid function, this can be resolved by using $\sigma$.
3. Sec 3.2 Para 8: "we randomly select $l$ instances and input them ..." $l$ has been previously used as layer index in Eqns 1-3, while $L$ has been used in Para 2 Sec 3.2 for the number of parallel steering pairs. Can you resolve this notation mismatch?
4. Equation 4: What is the dimensionality of $\phi^i$s? You could assume $N$ tokens, each represented by a vector in $\mathbb{R}^d$, as is typically done in self-attention mechanisms. Do you compute a $z^i$ corresponding to each position $i$? Is this computed differently for each attention head separately or is applied to the final output from all heads?
5. Why are there $\alpha_{prob}$ and $\alpha_{contr}$ two different hyper-parameters in the method? I believe a similar effect can be achieved just by scaling one of the two.
6. I could not clearly follow how the operation described in Eq 6 is applied during inference. Autoregressive decoding from an LM occurs one token at a time. Say you're predicting the $i$-th token, would the method then involve modifying as per Eq 6 the representations of all previously generated tokens at all layers (only MHSA output) to predict the $i$-th token? Modifying $a_i^l$ would also modify $m_i^l$ (Eqn 2 & 3).
7. Sec 4.1: Are any tests of statistical significance conducted to verify the effectiveness of detoxification on these 1k prompts and their responses as per DeStein?
8. Sec 4.1: How are the baselines implemented or are these results reported from prior work? Does $l$=20 here refer to layers or instances?


Minor writing issues/typos:
Para 1: No space between "(LLMs)" and the citation "(OpenAI ..". This is a recurring issue throughout the paper with all the references.
Sec 3.1 Para 1: We focuses --> focus
Sec 3.2 Para 1: counterfactuals --> counterfactual
Sec 3.2 Para 1: should be "Linear Representation Hypothesis holds (Park et al. (2023))" (space and brackets)
Sec 3.2 Para 1: should be "Park et al., 2023 demonstrated ..." (no brackets)
Sec 3.2 Para 3: the four steps should be captialized for consistency with the paragraphs below (e.g. "Unconditional Generation" instead of "unconditional generation").
Page 5 last para: predicated on Probing --> probing; that exhibit --> exhibits

**Reasons To Accept:**

- DeStein is free of training overhead, making it a possible solution for an important problem of detoxifying text generated by Language Models.
- Authors provide a large set of experiments that covers GPT-2, Llama-2, OPT and MPT models, and several relevant baselines like LMA, SELF-DETOXIFY, GeDi, DExperts etc. They also report metrics like diversity of generated text beyond simply fluency to better assess LM's linguistic quality.

**Reasons To Reject:**

1. The proposed approach is not sufficiently investigated or motivated - why is the merging/integration only applied on attention head activations and not on other ones e.g. FFN activations? There is no ablation study conducted to justify the effectiveness of this choice.

2. Writing issues, particularly with the notations used make it difficult for the reader to follow the paper. See questions 2 & 3 below. Fig 1 is hard to understand, and doesn't convey the method well.

3. The authors don't mention limitations like the memory footprint required to store the fixed/universal steering pairs for different models. This requirement scales with model size (n_layers). See question 4 below.

4. The scope of evaluation is limited to open ended text generation from GPT-2 style language models. While preserving linguistic quality of text generated from an LM is a good metric to assess a detoxification technique, in the context of present research on language models, the method could be made a lot more convincing by demonstrating its effectiveness by detoxifying models like Llama/OPT/MPT/Phi-2 while preserving their performance on benchmarks that cover language reasoning and understanding tasks. The universal steering proposed by the author should be verified to provide detoxification without hurting performance of LMs on practical tasks beyond open ended text generation.

---

> ### Author Rebuttal · Authors · 2024-05-31
>
> Thank you for your thoughtful comments!
>
> **Responses to the reasons to reject**
> - R1:Following your suggestion,we added ablation experiments applying activation fusion at FNN,MHSA,and FNN+MHSA.MHSA performed best,supporting our hypothesis.
>   ||Exp.Max.Tox.|PPL|
>   |-|-|-|
>   |FFN|0.404|148.14|
>   |MHSA+FFN|0.249|59.848|
>   |MHSA|0.207|39.434|
> - R2:Thanks for your suggestion.We apologize for any confusion caused.We'll include explanations of the notations in our revision.
> - R3:Following your suggestion,we discussed the limitations.
>   |Model|GPT2-L|GPT2-XL|LLaMA2-7B|
>   |-|-|-|-|
>   |TM|180KB|300KB|512KB|
>
>   $\text{Total Memory(TM)} = N\_l×N\_h×D\_h×B$,where $N\_{l}$  is layers, $N\_{h}$ is attention heads per layer,$D\_h$  is output vector dimension per head,and  $B$ is bytes per value (4 for float32).
> - R4:We added the MMLU dataset for validation in common evaluation tasks.Results show our method detoxifies well with little impact on general use.
>   |Model|Method|Ave.Acc|
>   |-|-|-|
>   |GPT2-L|base|0.27|
>   ||DAPT|0.237|
>   ||OURs|0.261|
>   |LLaMA2-7B|base|0.46|
>   ||DAPT(LoRA)|0.444|
>   ||OURs|0.459|
>
> **Responses to the questions**
> - A1: Both don't require changing the model's weights and affect model activations.Therefore,we categorize DESTEIN as a prompt-based variant.
> - A2: Yes,P and C denote prompt and completion.We apologize for any confusion caused.We'll change the symbol from $p$ to $\sigma$ for clarity.
> - A3: We'll use $l$ to denote the layer index and $m$ to denote the number of parallel pairs to ensure clarity.
> - A4: The dimensionality of $\phi^i$ matches the output size of one attention head. Each head $i$ computes a separate $z^i$ individually.
> - A5: Actually,$\alpha_{prob}$ is calculated based on accuracy.$\alpha_{contr}$ controls detox strength and can be set manually during inference (higher values mean more intensity).
> - A6: Yes. In the implementation,we achieve this by adding a simple additive computation network at the activation positions.
> - A7: Yes,we use 'Exp.Max.Tox.' by generating 25 responses per prompt and averaging their toxicity to statistically demonstrate DeStein's effectiveness.
> - A8: Baseline results were re-run with optimal parameters from the original papers.Details are in Appendix B.Here,$l=20$ refers to instances rather than layers.
>
> **Others**
> - We'll address the writing errors. Thanks for your thorough reading.
> - Due to word limits,only key results are presented.Full results,analysis,and citations will be included in our revision.

---

> > ### Comment · Reviewer_bowP · 2024-06-06
> > **Response to rebuttal**
> >
> > I thank the authors for the responses in their rebuttal. I find most of the concerns addressed from my review, and therefore am revising my score accordingly.

---

> > > ### Author Response · Authors · 2024-06-06
> > >
> > > Thank you for raising your evaluation score. We truly appreciate it! And thank you for the thoughtful suggestions to improve the paper, both regarding the discussion of the methodology and in improving the experimental section.

---

### Decision · Program_Chairs · 2024-07-10

**Decision:**

Accept

**Comment:**

This paper proposes an inference-time intervention approach to reduce the toxicity of model generations. While the approach seems somewhat incremental compared to various works on steering vectors, it is more lightweight than the baselines, and the experimental results show meaningful improvements. The discussion period has been productive, and the constructive feedback from the reviewers and engagement from the authors have led to significant improvements in the paper.

I’d like to highlight that while the authors show inspiration from Turner et al. (2023) and Park et al. (2023), there is no mention of more recent work that uses paired samples for generating steering vectors, such as "Steering Llama 2 via Contrastive Activation Addition" (Rimsky et al., 2024) or activation intervention such as "In-context Vectors: Making In-Context Learning More Effective and Controllable Through Latent Space Steering" (Liu et al., 2024). The latter particularly focuses on steering attention output, and the application is reducing toxicity. Better positioning with respect to these works, even if mentioned as “concurrent work,” would further improve the draft.